# Representation of polynyas in the Ross Sea coupled atmosphere—sea ice—ocean model P-SKRIPSv2

Alexandra Gossart<sup>1</sup>, Alena Malyarenko<sup>2</sup>, Liv Cornelissen<sup>3,4</sup>, Craig Stevens<sup>3,4</sup>, Una Miller<sup>5</sup>, Christopher J. Zappa<sup>6</sup>, Nancy Lucà<sup>7,9</sup>, Pasquale Castagno<sup>8</sup>, and Giorgio Budillon<sup>9</sup>

**Correspondence:** Alexandra Gossart (alexandra.gossart@vuw.ac.nz)

**Abstract.** Antarctic coastal polynyas are regions of intense air-sea exchange, strongly influenced by atmospheric winds, and significantly impacting sea ice formation and high salinity shelf water production. These areas play a key role in local and global climate feedback mechanisms. Due to their remote location and harsh conditions, direct sampling of polynyas is challenging, and capturing their dynamics in coupled climate models is difficult, especially when such models operate at coarse spatial resolutions.

In this study, we employ the regional P-SRKIPSv2 coupled atmosphere—sea ice—ocean model to simulate polynya activity in the Ross Sea. We evaluate the ability of the model to represent atmospheric and oceanic states and explore the sensitivity of the results to variations in the air-sea ice drag coefficient. Our findings demonstrate that P-SRKIPSv2 is a robust and effective tool for investigating polynya processes, including their drivers and impacts, in the Ross Sea region.

#### 10 1 Introduction

Coastal, or latent heat, polynyas are found along the Antarctic coastline in winter, where the sea ice intermittently opens under the action of strong winds blowing over the sea ice (Bromwich and Kurtz, 1984; Kurtz and Bromwich, 1985). These offshore winds break the sea ice surface and push it away from the coastline for prolonged periods of time (Morales-Maqueda et al., 2004). Polynyas have typical length scales of order 100 km, vary in size throughout the austral winter, and are recurrent over many years (Morales-Maqueda et al., 2004; Martin, 2019). The intense offshore downslope/katabatic winds and the exchanges of heat and moisture at the exposed surface of the ocean initiate the production of new sea ice (Kurtz and Bromwich,

<sup>&</sup>lt;sup>1</sup>Te Puna Pātiotio | Antarctic Research Centre, Te Herenga Waka | Victoria University of Wellington, Te Whanganui a Tara | Wellington, Aotearoa | New Zealand

<sup>&</sup>lt;sup>2</sup>Te Kura Aronukurangi | School of Earth and Environment, Te Rāngai Pūtaiao | College of Science, Whare Wānanga o Waitaha | University of Canterbury, Ōtautahi | Christchurch, Aotearoa | New Zealand

<sup>&</sup>lt;sup>3</sup>Ocean Physics Group, Earth Sciences New Zealand, Te Whanganui a Tara | Wellington, Aotearoa | New Zealand

<sup>&</sup>lt;sup>4</sup>Te Kura Mātai Ahupūngao | Department of Physics, Waipapa Taumata Rau | University of Auckland, Tāmaki Makaurau | Auckland, Aotearoa | New Zealand

<sup>&</sup>lt;sup>5</sup>Department of Earth and Atmospheric Sciences, Cornell University, Ithaca, NY, USA

<sup>&</sup>lt;sup>6</sup>Lamont Doherty Earth Observatory of Columbia University, Palisades, NY, USA

<sup>&</sup>lt;sup>7</sup>Department of Environmental Sciences, Informatics and Statistics, Ca' Foscari University, 30172, Venice Mestre, Italy

<sup>&</sup>lt;sup>8</sup>Department of Mathematics and Computer Science, Physics and Earth Sciences, University of Messina, 98166, Messina, Italy

<sup>&</sup>lt;sup>9</sup>Department of Science and Technology, "Parthenope" University of Naples, 80143, Naples, Italy

1983, 1985; Tamura et al., 2008), which in turn leads to the formation of dense High Salinity Shelf Water (HSSW) (e.g., Budillon et al., 2003; Fusco et al., 2009; Rusciano et al., 2013), a precursor of the Antarctic Bottom Water (Silvano et al., 2023). These waters are part of the global thermohaline circulation and have an impact on the ventilation of the deep ocean (Gordon et al., 2009).

The fraction of open ocean constitutes a local source of heat and moisture for the atmosphere, impacting the local mesoscale circulation (Batrak and Müller, 2018). A modelling study of two way air-sea interaction in the Terra Nova Bay Polynya (TNBP) and Ross Sea Polynya (RSP) (Jourdain et al., 2011) shows that heat and moisture fluxes to the atmosphere are affected by the reduction in sea ice cover, in turn negatively impacting sea ice production and deep water formation. This highlights the need to consider the polynya air-sea ice-ocean system and its feedbacks as a whole, in order to assess the responses of the different components to intra- and interannual variability as well as to long term changes of the climate system.

Moreover, the sea ice cover drifts under the action of strong winds, which often leads to a shear between the sea ice and the ocean affected by the currents and tides. The difference between the currents in the ocean and the sea ice movement generates turbulence, which is also controlling the effectiveness of the air-ocean exchanges. However, measurements under sea ice, especially boundary layer thickness and friction velocity, are scarce to non-existent for the Antarctic. Based on a handful of observations and estimates of velocities in the sea ice-ocean boundary layer, a large range of drag coefficient values have been proposed: Fer et al. (2022) found values between 4 and  $6.10^{-3}$  from measurements taken in the Arctic during the MOSAIC expedition. In the Antarctic, values between  $0.13 \cdot 10^{-3}$  and  $47 \cdot 10^{-3}$  have been reported between 1970 and 1997 (Steele et al., 1989; Lu et al., 2011). More recently, the values proposed range from 1.1 and  $5.36 \cdot 10^{-3}$  (Kim et al., 2006; Massonnet et al., 2011; Stopa et al., 2018; Wamser and Martinson, 1993), and even reach 0.87 to  $5.5 \cdot 10^{-3}$  in Schroeter and Sandery (2022). Because of the difficulties in measuring below sea ice in both the Arctic and Antarctic, in climate models, set drag coefficients (fixed in space and time) are used in order to represent the dynamical processes of sea ice induced by the wind and ocean currents through momentum exchange in the horizontal direction (Lu et al., 2011). This implies that large uncertainties remain in the estimation of the rate of exchange between the ocean and the atmosphere for heat, mass and momentum transfers.

In the context of climate change and future climate projections, General Circulation Models (GCMs) are used to predict the future state of the climate in Antarctica. In general the CMIP6 models (Eyring et al., 2016) represent an ocean surface that is too warm and too fresh for the Southern Ocean. The annual sea ice extent is also too small (Beadling et al., 2020) and there is a wide spread in the different terms of sea ice formation/dissipation, leading to large uncertainties in sea ice budget across the different models ((Li et al., 2021). GCMs in general struggle to represent realistic polynya extent and variability (Mohrmann et al., 2021).

Higher resolution studies with coupled ocean-atmosphere-sea ice regional models are possible (e.g., Jourdain et al., 2011), but remain quite scarce over the Antarctic coast and only a few are applied to polynyas. In those models, the correct representation of air-sea ice-ocean interactions is still a challenge, albeit crucial to be able to predict Antarctic sea ice variability in a changing climate. To address this gap, we investigate the representation of coastal polynyas in the fully coupled air-sea ice-ocean regional P-SKRIPS model version 2 (P-SKRIPSv2), an updated version of P-SKRIPSv1 (Malyarenko et al., 2023). This coupled model does not contain a dedicated sea ice model, but the sea ice is embedded in the ocean component. This model was specifically

developed to conserve heat and mass fluxes and allows for feedback between the ocean and atmosphere. It is applied to a Ross Sea domain that encompasses the RSP and TNBP. We assess the skill of the P-SKRIPSv2 model to simulate polynyas as well as the sea ice cover and thickness to satellite products and reanalyses over the year 2017. We also investigate how well the model simulates atmospheric drivers and the ocean response to the polynya extent. We further test the impact of varying the air-sea ice drag coefficient from  $1 \cdot 10^{-3}$  to  $3 \cdot 10^{-3}$  on sea ice and polynya opening.

#### 2 Methods

#### 2.1 Coupled model set up

We use the P-SKRIPSv2 model, an improved version of the model published as P-SKRIPSv1 (Malyarenko et al., 2023). This model uses the Massachusetts Institute of Technology Global Circulation Model (MITgcm) to simulate the ocean and dynamic/thermodynamic sea ice (Losch, 2008), and the polar version of the Weather Research and Forecasting Model (PWRF hereafter; Skamarock et al. (2008)) for the atmosphere. Both components are coupled using the Earth System Modeling Framework (ESMF). In the MITgcm model, the sea ice package is a simplified viscous-plastic sea ice model, including zero-layer thermodynamics with snow cover (Semtner, 1976). This routine allows the sea ice layer and snow on sea ice to evolve and grow, but considers that ice heat capacity is zero (sea ice does not store heat). In this version, the thickness category parameterization is set to 0 (sub-grid scale ice thickness distribution is ignored). Although the sea ice routine is limited, this set-up enables the conservation of the mass and heat fluxes exchanged between the two models. It is applied over the Ross Sea domain (Figure 1) at 10 km horizontal grid spacing, has 61 vertical levels in the atmosphere and 70 levels in the ocean. The models exchange information every minute, and the timestep of the ocean and atmosphere are 60 and 20 seconds, respectively.

The first improvement to P-SKRIPSv1 concerns the implementation of the exchange of the thickness of the snow layer on top of the sea ice between the ocean and the atmosphere models. This allows the atmosphere to impact the snowpack on top of sea ice via accumulation, metamorphism, and snow compaction processes; the ocean model is responsible for the advection of the snow cover on top of the sea ice across the grid cells, and sends the updated sea ice and snow on sea ice fields back to the atmosphere component. The second improvement relates to an homogenisation of the definition of two variables: the thickness of the sea ice and the thickness of the snow layer on top of that sea ice layer. These two variables are defined differently in the atmosphere and ocean components, thus requiring adaptations in the exchange procedure to reflect these dissimilarities. The current version of the code (P-SKRIPSv2) can be found at: git@github.com:alenamalyarenko/pskrips and on zenodo: 10.5281/zenodo.15942874.

#### 2.2 Coupled model experiments

We investigate the effect of varying the sea ice drag coefficient at the air-sea ice interface in the ocean component of the coupled model. This coefficient is parametrized by a fixed number in the MITgcm settings (SEAICE\_drag in data.seaice) and is usually set to  $1.0 \cdot 10^{-03}$  (MITgcm users manual). We run a series of simulations, varying the drag coefficient to  $2.0 \cdot 10^{-03}$  (sim 002)

Figure 1. Simulated domain, centered over the Ross Sea - red square in a) -b) extent of the model domain, the shading corresponds to the land elevation in the atmosphere model, and the sea ice cover over the ocean (note the lighter blues and white hues at the location of the polynyas). The two weather stations investigated in this paper are indicated in white (Manuela and Vito) and the boxes defining the Terra Nova Bay (TNBP) and Ross sea (RSP) polynyas as well as the open ocean (OO) are delimited in red. The yellow dots represent the voyage track of the RVIB Nathaniel Palmer in the Ross Sea. c) zoom onto the TNBP area - background is same as in b)-, the red box delineates the TNBP area used in this study. The white dot indicates the location of the Manuela automatic weather station, the black dot represents the grid cell from which the wind speed was extracted for the flux investigation. The yellow stars indicate the location of the two moorings DITN and DITD along the Drygalski Ice Tongue, the purple star indicates the LDEO mooring and the green star is the MORSea D mooring.

and  $3.0 \cdot 10^{-03}$  (sim 003) as well as a default simulation using the standard  $1.0 \cdot 10^{-3}$  (sim 001) value. In the coupled model, the SEAICE\_drag is only affecting the advection scheme in MITgcm, and is not part of the bulk thermal flux parameterisation, which implies that this change generates dynamic and thermodynamic variations in simulated sea ice variables, but the thermal fluxes are not impacted as they are provided by PWRF directly. These imported atmospheric fluxes remain calculated using the air-surface drag coefficient based on the boundary layer theory in the PWRF land module Noah-MP (Yang et al., 2011).

Our analysis focuses on 3 distinct areas in the modelled domain, visible as the red boxes in 1b): one region located away from the coast and representing the 'open ocean' (OO): this area is totally free of sea ice in summer and fully covered during the winter. The second area encompasses the TNBP and extends along the 70 km of the Drygalski Ice Tongue (DIT) and up to 75° South. The third domain covers the RSP: along the coast of the Ross Ice Shelf and from 170° East to 180°. Both the RSP and TNBP are reported as areas of active sea ice production in winter (Kurtz and Bromwich, 1985; Dai et al., 2020; Hollands and Dierking, 2016), and are characterised by open water or reduced sea ice cover, surrounded by sea ice.

# 2.3 Observational Datasets

We compare the performance of the coupled model simulations to a series of model outputs and observations (for more details, see section S1 and Table 1). We use the satellite-derived datasets of sea ice cover (SIC) and sea ice thickness (SIT) for the OO, RSP and TNBP areas. The various products are listed in Table 1 and have daily or monthly temporal resolution, and range from a few km (3.125 km) to 25 km grid spacing. We further compare the simulations to available reanalysis products (ORAs5 and Glorys) and the Antarctic WRF Mesoscale Prediction System (AMPS) output and finally to the World Ocean Atlas (WOA). For all these gridded datasets, a masking of the coast and regridding to P-SKRIPS spatial grid are carried out before extracting the different areas and computing the spatial and temporal means. We also use direct measurements of ocean temperature and salinity and compare them to the closest model grid cell. Several moorings have been deployed in the TNBP area: DITx (yellow start is Figure 1c)), Lamont-Doherty Earth Observatory (LDEO, purple stars in Figure 1c)), and the MORSea mooring D (green star in Figure 1c)). The vertical sections of the ocean salinity and temperature are extracted along a line following the ice shelf front, which varies according to the datasets (due to the different spatial resolutions, see Figure S1). The vertical profiles are extracted at 74.982155 °S and 165.46388 °E (red star in Figure 1 c)). We compare the closest grid cell of modelled meteorological variables with daily observations from Automatic Weather Stations (AWS). For this purpose, we select the stations located upstream and closest to the two polynyas—Manuela (74.946 ° S, 163.687 ° E) for the TNBP and Vito (78.408 ° S, 177.829 ° E) for the RSP (white dots in Figure 1 b). These sites also represent contrasting environments: one situated in the complex terrain of the Transantarctic Mountains (TAM) and upstream of the TNBP, and the other on the edge of the Ross Ice Shelf where the winds blow offshore and open up the RSP. This contrast allows us to assess model performance across different geographical settings. We also contrast the model with the measurements taken onboard the vessel during the PIPERS cruise (Polynyas and Ice Production in the Ross Sea, yellow dots in Figure 1 b)) at the closest model hourly output. For all, we correct the temperature by applying a dry adiabatic lapse rate of  $9.8\,^{\circ}\text{C.}km^{-1}$  (Bromwich and Fogt, 2004) to account for discrepancies between the modelled grid cell elevation and the actual station elevation. In addition,

**Figure 2.** Evolution of the daily (monthly) SIC as a fraction over time (01-02-2017 - 31-12-2017) for the TNBP domain, the RSP domain and the OO domain. SIC ranges from 'fully covered by sea ice' (1) to 'fully open ocean' (0) and 'partially open sea ice/polynya' is identified as having a SIC around 0.6-0.8. The vertical lines indicate the 15th of each month. Due to the masking and resolution of the dataset, the NOAA product only has information for the OO box.

we extrapolate the observed wind speed from 2m to 10 m according to Sanz Rodrigo (2011). The wind time series in Figure 4 is extracted from an ocean grid cell just north of the DIT (163.82304 ° E, 75.14886 ° S).

# 3 Results

#### 3.1 Sea Ice


# 3.1.1 Sea ice cover and polynya area

In general, both the observations and model outputs display a similar yearly sea ice cycle in Figure 2: The three domains are virtually free of sea ice at the model initialisation stage, followed by an onset of sea ice formation in early to mid-March (mid-February for AMSR), and reach the first maximum sea ice concentration (SIC) within the domains in early April (or early March for AMSR). In March through to May, all products and simulations show a rapid increase in SIC (over 2 to 3 weeks) to the mean winter values. The sea ice cover then fluctuates around its maximum until the break-up season (mid-November in all locations).

All model simulations display a build up of SIC 2 to 4 weeks later in the season than the satellite-derived data. The sea ice break up also starts systematically around a month earlier in the model simulation but reaches a minimum SIC at the same time








as the satellites: close to 0 in the polynyas for all by mid-November to early December, except Bootstrap, AMSR and Cryosat (and Glorys in TNBP that show a re-increase in December). In the OO box, however, all satellite products and reanalyses indicate some presence of sea ice when the model displays none in December. The observations of yearly mean SIC are around 0.6 in the TNBP area, 0.5 in the RSP and 0.8 in the OO (Table S1). The simulations display a lower yearly mean SIC than the satellite products, with a mean 0.3 in the polynyas and around 0.5 in the OO. While not as low, ORA and SSMI have the lowest values over the polynya areas. Overall, sim 003 has the lowest SIC for the three areas and over the whole time period but is in general closer to the values of sim 002, while sim 001 constantly displays higher SIC. In TNBP, sim 001 aligns best to the observations (RMSE). Along the RSP, sim 002 and 003 SIC are systematically under the satellite-derived products and sim 001, but are closer to the Glorys and ORA reanalyses. AMSR SIC is close to 1 during the whole year, which indicates that no polynya is observed in the Ross Sea by this product. All models and observations agree on a nearly fully covered OO during the winter season. The OO SIC tends to be underestimated in the model outputs: in march-may values of 0.6 to 0.7 SIC are observed while the simulated mean is around 0.5. The variability of the SIC, however, is consistent between the observations and the model results.

The results indicate that the seasonal cycle of SIC is well captured by P-SKRIPS: the polynya areas show variability in SIC of similar magnitude to the observations. The RSP is especially well matched by the models, and TNBP is a bit more challenging to simulate as it is strongly affected by the local conditions and topography, and counts only a few grid cells in the P-SKRIPS model. The differences over the 'OO' box are mainly attributed to the boundary conditions: the amount and thickness of sea ice advected at the borders of the model which suffer from inaccuracies and uncertainties.

## 3.1.2 Sea ice thickness

Figure 3 displays the mean thickness of the sea ice in the three areas from satellite-derived datasets, reanalyses and simulated sea ice thicknesses from the different models.

The satellite-derived thickness products (CryoSat and SMOS) show large differences in sea ice thickness (SIT), with CryoSat generally providing the upper estimates and SMOS the lower estimates for both polynya areas. However, they agree on values centered around 1 m in the OO (Table S2). Rack et al. (2021) reported a mean SIT of  $2.0\pm1.6$  m (with a maximum of 15.6 m) in the western Ross Sea during a November 2017 campaign. These values may suggest that CryoSat estimates are closer to reality than those of SMOS, indicating the potential for thicker sea ice in the region. In TNBP, the SIT is slightly larger in the reanalyses, until the beginning of October, when it suddenly increases in all products. The SIT peaks at over 1 m then, after having oscillated around 0.5 m from mid-March onward. While with a different magnitude, this variation in SIT is visible in the satellite products also. Within the RSP box, the SIT does not vary much and remains around 0.5 m thick for the whole year (except for a slight increase at the end of the simulations). Satellite products also show insufficient variation over the year. Finally, the OO area shows agreement between the two satellite products. Both display values around 1 m thickness, while the reanalyses and simulations only reach 0.5 m SIT and show little changes throughout the year. Except for the months of July and August, model outputs and reanalyses SIT lay very close to each other and sim 002 is closest to the reanalyses.



**Figure 3.** Evolution of the daily (monthly) SIT [m] over time (01-02-2017 -31-12-2017) for the TNBP domain, the RSP domain and the OO domain. The vertical lines indicate the 15th of each month.

# 3.2 Polynyas: drivers and impacts in the TNBP

P-SKRIPS successfully models three typical types of atmospheric events, and the resulting variations in the extent of TNBP (Figure 4): 1) conditions similar to an atmospheric river at the end of April; 2) a traditional 'katabatic event', mid-July; and 3) the closing of the polynya due to an onshore wind, beginning of October.

The intrusion of warm and moist air (AR) was reported by Fonseca et al. (2023) and is related to an increase in TNB polynya size between 27-29th of April 2017. This event is highlighted in blue (first section). During these 3 days, the wind speed and its U component are low and below the  $20 \ m.s^{-1}$  threshold defined by Aulicino et al. (2018) for a wind event to impact the extent of the polynya. The long wave radiation flux increases during the time period, due to the presence of moist and cloudy weather, while the sensible and latent heat fluxes are close to zero. The 2 m temperature and humidity see a peak during the event. Despite the weaker winds, the polynya is widening due to the intrusion of warm air. The SIC displays a dip to  $\sim$ 0.4, a reduction to below the 0.6 and 0.7 thresholds defined by Burada et al. (2023) and Aulicino et al. (2018). The ocean temperature decreases at the time of the event, indicating heat loss to the atmosphere, and the event is followed by sea ice production, shown by increase in SIC, ocean salinity and surface temperature in the next days.




A second type of event is visible in the graph (second and third highlighted sections) between the 12th and 20th of July 175 (and also 26th of July until 16th of August). During the 12th-20th of July, the wind speed shows a large increase and reach 25  $m.s^{-1}$  - and even a peak of 30  $m.s^{-1}$  in the zonal wind component- and sustained wind speed over 20  $m.s^{-1}$  remain over the following days. This coincides with a maximum in latent and sensible heat fluxes and a minimum in long wave radiation. The humidity and temperature are low - indicating the presence of dry and cold air; and the SIC decreases and crosses the polynya threshold. As strong cold winds expand the polynya area, new sea ice forms, a process reflected in the salinity timeseries by a 180 noticeable increase during and after the event. In this example of a katabatic event, the simulated wind is westerly and largely oriented along the U wind component. The katabatic winds are characterised by clear skies (reduced long wave radiation) and are cold, dry winds: the humidity and temperature drop at the beginning of the event. In addition, the presence of a polynya promotes larger ocean heat exchange to the atmosphere, leading to local cooling and sea ice production. The strong offshore 185 winds blow from the Transantarctit Mountains and open up the TNBP, the air-sea exchanges are enhanced over the open water, increasing the temperature, turbulence and moisture transport (Guest, 2021). Both the latent and sensible heat fluxes are impacted as evaporation, cooling and sea ice formation occur. The thickness and presence of sea ice are modulated by the wind, and in turn regulate the latent and sensible heat fluxes, which increase as the SIC reduces and decrease as the sea ice gets thicker. This event is one example of the man katabatic events occurring through the winter.

Finally, the event lasting from the 3rd until 6th of October is characterised by strong winds in the TNBP, but mainly directed onshore (this is visible in Figure 5as the large dip at the beginning of the month). The heat and radiation fluxes show a behaviour similar to the April event, and so do the humidity and temperature curves. On the contrary, the SIC and SIT display (large) increases in the TNBP area during this event, altering oceanic conditions and leading to a decrease in salinity, likely reflecting the advection of different water masses into the TNBP box. This third event captures a reduction/closing of the TNBP area as well as a piling up/thickening of sea ice under the influence of onshore winds (Figures S2 and S3). Signatures of a storm passing through the Ross Sea are present in Era-5 (not shown): a low pressure system in the region, sharp changes in wind direction and speed as well as increased cloud cover and snowfall making landfall in the TNBP area.

Large variability in long wave radiation, sensible, and latent heat fluxes is evident in Figure 4; these heat fluxes have not been measured in Antarctic coastal polynya areas, except by Knuth and Cassano (2014) and Guest (2021). They report 12 to 485  $W.m^{-2}$  for sensible heat - 56-152  $W.m^{-2}$  for latent heat; and 2500 +-600  $W.m^{-2}$  for sensible heat and 400  $\pm$ 100 for latent heat and  $100\pm7~W.m^{-2}$  for radiation heat fluxes, respectively. Both studies used local temperature, pressure, humidity and wind and to estimate these fluxes, which are associated with large uncertainties due to the choice of heat transfer coefficients (Miller et al., 2024a). The results of our coupled model are closest to Guest (2021): We observe that SH is dominating the total surface flux and long wave radiation is the smallest contributor. As temperature and wind control heat fluxes, sensible and latent heat fluxes show very similar behaviors (Guest, 2021).

# 3.3 Atmospheric variables in TNBP

Of greatest importance to polynya initialisation and persistence is the accurate representation of the U and V components of the wind. On the one hand, their correct representation is challenged by the complex topography, as the winds are channeled

radiation (Iwnet), latent heat flux (lh), sensible heat flux (sh), 2 m temperature (2m t), 2m relative humidity (2m rh) 2m specific humidity (2m q), sea ice cover Figure 4. PSKRIPS hourly atmospheric variables from sim 002: 10 m wind speed (10 m ws), 10 m U-component of the wind speed (10 m u), net long wave (sic) and sea ice thickness (sit) and oceanic variables: ocean surface temperature (tos) and salinity (salt). Mean over the TNBP box. The shaded areas represent the duration of the three events highlighted in this section.






through the TAM. These subtle effects might be missed in the coarser model simulation. On the other hand, the climate of the Ross Ice Shelf is modulated by local and synoptic conditions, and the combination of katabatic winds and barrier winds along the TAM (Parish and Bromwich, 1997) form the Ross Air Stream (Costanza et al., 2016), known to cause sea ice breakup in the RSP (Dale et al., 2020) and these specific conditions could make it challenging for the models to reproduce correctly.

In general, the observed meteorological data and their representation in atmosphere models are fairly close for all variables at both Manuela and Vito stations (Figures 5,6 and Figures S4 and S5, and Table 2). There is a quasi-overlap between AMPS, P-SKRIPS and the AWS observations: the mean state as well as the variability are well matched at the two locations. The variables displaying the largest RMSE to the observations vary between the stations, indicating no spatially consistent bias of the models, which is also reflected in the WI. In this section, we focus on Manuela AWS and results for the Vito station are detailed in Section S2.

Manuela station, located upstream of the TNBP in complex terrain, is more difficult to simulate than Vito. Overall, both AMPS and P-SKRIPS display a similar behaviour for all meteorological variables, except relative humidity. It is clear that there is a constant overestimation of the relative humidity by AMPS (least WI for that station, 0.36), at least until the month of October when all time series converge (Figure 5)). Overall, P-SKRIPS matches the AWS fluctuations better, albeit displays a near-constant underestimation (WI of 0.53). The shape of the distribution of the observed data is reproduced by P-SKRIPS, but with a peak at ~30% while the model's reaches around 45% (Figure 6). On the other hand, AMPS shows a narrower distribution and is skewed towards larger values with a peak at around 65-70% and no values below 40%, in contrary to the other datasets. Both models near-perfectly follow the temperature curve, exhibiting the same fluctuations as the AWS measurements. The 2m temperature is well matched by the models, which lies very close to the observations. Overall, the wind speed shows a very good agreement both between the two model outputs, and with the observational data. The fall months of June and August seem to be the months driving the offset towards higher wind speeds for the models, as the magnitude of the rest of the year is well matched by the simulations (WI of 0.65 and 0.73 for P-SKRIPS and AMPS, respectively). In Figure 6, both models are skewed to the right (overestimate) wind speed with a peak at around 30  $m.s^{-1}$  while AWS measurements show maxima at 5 and around 15  $m.s^{-1}$  and only a few occurrences of wind speeds above 25  $m.s^{-1}$ . P-SKRIPS displays the largest difference to the AWS data (Table 2), which is the case at both stations. The direction of the wind is well followed by P-SKRIPS and AMPS: most of the sharp changes in the observations are also present in the model outputs. The predominance of Westerly-Southwesterly winds is well matched by P-SKRIPS and in this case, AMPS performs a little less well (RMSE of 36.24 and 43.09, respectively). Finally, there is a clear  $\sim 20$  hPa offset in the modelled surface pressure at Manuela, leading to the lowest WI for this location (0.34 - P-SKRIPS and 0.38 - AMPS). The variability in the observations, however, is well followed by the models, indicating a good representation of the general pattern.

Our simulations are able to match the automatic weather station measurements over land at Manuela and Vito stations. In addition, the surface temperature, wind speed and direction and the pressure are also well represented in P-SKRIPSv2 over the ocean surface (section S2 and Figure S6).

**Figure 5.** a) Time series of five meteorological variables at Manuela station (78 m asl), February until December 2017, P-SKRIPS sim 002 closest grid cell (0.26 km, 12 m asl, khaki), AMPS model grid cell (blue) and AWS data (pink).

**Figure 6.** same data as in Figure 5, but in the form of distribution. For simplicity, we show only one model simulation (sim 002, khaki) as all the model outputs are very close to each other, given that the variation of the drag coefficient impact locations offshore, downstream of the stations. The AMPS dataset is in pink and the AWS data is in blue. The vertical lines denote the 15th of each month.

# 3.4 Atmosphere-ocean connection in TNBP




Figures 7 and 8 show the vertical profiles taken from the PIPERS ship, using rawinsondes to sample the temperature and wind in the atmosphere (Figure 7) and CTDs to sample the ocean salinity and temperature (Figure 8). We compare these to the model outputs and note that our simulations represent the right vertical structures in both the atmosphere and the ocean components of P-SKRIPS. The atmosphere models match well the vertical profiles of the measured temperature (first column) and AMPS (third column). The slight temperature inversion in the first profiles is present in all three graphs (blue and green lines, first row), but this layer is shallower in the AMPS dataset. It also corresponds to the top of the katabatic layer (*U* component of wind, second row in Figure 6). Over time, the inversion disappears, disrupted by the winds, to a near linear profile (yellow lines). This behaviour is slightly better matched by AMPS than P-SKRIPS, although it is present in both simulations.

The shape of the temperature and salinity profiles with depth in the ocean display similar shapes for P-SKRIPS (top) and PIPERS data (bottom) in Figure 8, although earlier in time in the model output: the salinity stratification with limited vertical convection (around 200m, blue shades) increases with time, and reaches 600 m depth for both datasets by the end of the period (yellow shades). We also note the convergence of the measurements at depth in the model outputs, which corresponds to the curves of the measured variables. The observed temperature profile displays the presence of a warm core at 200m depth

Figure 7. Vertical profiles of the atmosphere taken onboard the PIPERS ship (left column), corresponding data from the P-SKRIPS model (central column) and AMPS (right column). The first line displays the temperature, the second and third show the U and V wind components, respectively. Each profile is displayed with its elevation and the color represents the date at which the profile was sampled.


**Figure 8.** CTD profiles of the ocean taken from the PIPERS ship (bottom row), corresponding data from the P-SKRIPS model simulation 002 (top row). The first column displays the salinity profiles, the second shows the ocean temperature. Each profile is displayed with its own depth and the color represents the date at which the profile was sampled.

at the beginning of the CTDs, and this is also visible in the model results. The lack of cold Ice Shelf Water around 600m in the simulations is related to the removal of a nearby cavity in the model domain. The progressive mixing of the salinity and temperature through depth is a result of the strong winds blowing at the surface and illustrate the ability of the model to simulate the connectivity between the atmosphere and ocean components. Increasing the drag coefficient leads to a quick vertical mixing of the whole water column, resulting in a homogeneous profile with colder and saltier water produced in less than a week (Figure S7, simulation 003). All simulations produce a deep convection event in late April 2017 (roughly 2 weeks earlier than observed), corresponding to a local wind speed maximum (Figure 4, *U* component of the wind speed immediately before the April heat wave signature; Figure 5, wind speed at Manuela AWS location).

# 3.5 Comparison with hydrographic moorings in TNBP

Figures 9, 10 and 11 show the temperature and salinity timeseries of the model output compared to the hydrographic moorings in the TNBP: DITx, LDEO and mooring D respectively. The model captures the seasonal cycle of temperature and salinity

**Figure 9.** Temperature (left) and salinity (right) measured by the DITx moorings, DITN (pink) is the long mooring close to the Drygalski Ice Tongue (DIT) and DITD (orange) is the short bottom mooring in the Drygalski trough. The variability measured by both moorings between 2015-2020 is plotted in grey. Please note the y-axis limits are relaxed for visibility.

fairly well. The temperature increases in late summer; the lowest temperatures with the smallest variability are observed during the winter period. In addition, the delay in the temperature response with depth is similar to the mooring observations. The freshest water masses are observed near the surface in February, as in the mooring salinity observations. From the end of summer, the salinity slowly increases until it peaks by the end of winter, and decreases during spring.

During summer, the model systematically overestimates the temperature in the surface layers (DITx and LDEO, Figures 9 and 10), and shows a larger variability than the observed temperature. The model starts with a higher temperature in 2017 than measured, and also decreases earlier (Figure 9). However, compared to other years measured by the DITx, the peak in the mooring temperature was relatively late. The model is able to recreate the seasonal cycle of salinity at the DITx and LDEO locations (Figures 9 and 10), though it consistently overestimates measured wintertime salinity at all depths. As with temperature, the model exhibits the greatest bias in salinity during February through March.

**Figure 10.** Temperature (left) and salinity (right) measured by the LDEO moorings (pink). The model output is the corresponding cell in the PSKRIPS model. Please note the y-axis limits are relaxed for visibility.




**Figure 11.** Temperature (left) and salinity (right) measured by mooring D (pink). The model output is the corresponding cell in the PSKRIPS model. Please note the y-axis limits are relaxed for visibility.

The summer surface offset in the model is a known issue, also present in the spinup simulation. A fresh bias in the surface of the ocean allows for a stable stratification, and without sufficient mixing the summer surface layer is too shallow and too warm (the shortwave radiation does not mix down quickly). Then, as sea ice forms around March-April, the model and observations reconcile. The fresh bias originates from the ocean boundary conditions, and a higher air-sea ice drag coefficient was used in the spinup to reduce this artefact. However, that also caused a HSSW layer with overestimated salinity (Figure S8), see salty bias at depth in TNBP; and Figures 9 to 11 at deeper layers).

The end of the year sees a similar problem at the modelled surface, which might be related to a thick sea ice layer, which then produces too much freshwater when melting at the onset of the warm season (the October freshening signal visible in the mooring data matches early sea ice loss in the model, Figure 2). This aspect is also present in the reanalyses (Figures S8 and S9, Glorys). While these surface features are a model limitation, P-SKRIPS displays an improvement compared to the reanalyses (lies closer to the WOA values).

The salinity signal in the Mooring D illustrates the impact of the drag coefficient (Figure 11). The simulation with the highest drag coefficient results in larger cumulative salinity increase, starting earlier in the season. This allows for a lower freezing point, leading to a corresponding colder temperature and early cooling. Although the observed temperature and salinity at the







end of the year are similar to the beginning (pink lines in Figure 11), the model notably drifts to a colder and saltier state at the end of winter. The same signal is present at deeper layers at DITx (668 and 1225 m, Figure 9), but not at the shallow layers.

### 3.6 Ocean response along the RIS front

In this section, we investigate the salinity (Figure 12) and temperature (Figure 13) for the months of June and December 2017 (Section S3 and Figures S10 and S11) of the ocean along a transect following the front of the ice shelf, whose location differs between the different datasets due to the spatial resolution and the land-ocean mask used by the different products (see Figure S1). We compare the three model outputs (sim 001, 002 and 003) to the ORAs5 and Glorys reanalysis, and to the World Ocean Atlas dataset.

The East-West salinity gradient is visible in all datasets and model outputs (Figure 12): more saline HSSW in the Western Ross Sea (WRS) and a gradual appearance of Low Salinity Shelf Water (LSSW) in the Eastern Ross Sea (ERS). The P-SKRIPS simulations display an input of more saline waters from the surface and an interruption in the stratified layers at around 178° W - very faint for sim 001 but more marked as the drag coefficient increases. The ocean salinity is very stratified in the ORAs5 profile, leading to quasi horizontal isohalines and a near absence of the East-West gradient. The Glorys reanalysis exhibits a structure very similar to that of our simulations, with an input of salinity in the WRS, and fresher surface waters in the ERS. Finally, the WOA indicates a much more homogeneous salinity profile, with very different isohalines in an 'L' shape and lacks freshwater at the surface (especially in the ERS). The mean June temperature profiles (Figure 13) also show an East-West gradient: colder water in WRS than in the ERS, for all datasets. The coupled model simulations indicate the presence of warmer water entering the cavity in the ERS, leading to positive temperatures at depths of 300 - 600 m below the surface and reaching their maximum at the Eastern edge of the profile (circa 1°C). ORAs5 shows the same horizontal stratification as for salinity, with very warm waters sitting at the bottom, reaching 2°C in the troughs. Glorys and WOA profiles show less spatial variations in the profiles, no temperature is higher than 0°C, and both have an inflow of slightly warmer waters at depth in the ERS.

The P-SKRIPS simulations show expected patterns in the Ross Sea: modified Circumpolar Deep Water (mCDW) entering the cavity in the Central Ross Sea, as in Smethie and Jacobs (2005) and Jacobs and Giulivi (2010). However, the mCDW is expected to be cooler than -1 °C in the 2010s at the ice shelf front section (Jacobs and Giulivi, 2010) while the model simulation is about half a degree warmer. In addition, the modelled inflow of water (+0.5 °C) in the Little America Basin has not been observed. This could be attributed to the boundary conditions, providing an inflow of too warm waters in the ERS, or could be a feature that exists in the ERS, but has not been observed yet.

The atmospheric conditions drive the opening/closing of the polynya, which initiates changes in the ocean state: There is a clear polynya signal in the coupled model output, visible as vertical isohalines from the surface in Figure 12 (left panels) and reaching 200 to 400 m depth into the ocean at 178 ° W. Simulation 001 displays only a faint signal and the increase in salinity is not very visible, but simulations 002 and 003 have a much clearer signal (brighter color indicating larger salt content). The larger salt input at the surface of the ocean is a reflection of brine rejection occurring when sea ice forms, when the polynya is more active. As the sea ice forms, vertical convection in enhanced, particularly in the polynyas, creating a column of saltier and colder waters, HSSW. This is visible in the temperature section as a gap in the horizontal layer that sits between 200 and

Figure 12. Salinity profiles and contours along the ice shelf edge, monthly mean values for June 2017

400 m depth, at the 178 ° mark in the profiles in Figure 13; and as vertical isohalines at the same location in Figure 12. This signal is stronger with higher drag coefficients. In Figures S10 and S11, the signature of the polynya disappears as the oceanic waters stratify (Figure S10) and the surface waters are slightly warmer than for the rest of the profile (Figure S11), due to the fact that SIC is reduced in the polynya area, hence warming faster at the start of summer.

Figure 13. Temperature profiles and contours along the ice shelf edge, monthly mean values for June 2017.

#### 4 Discussion








#### 4.1 Air-sea ice drag coefficient

In this paper, in addition to examining the performance of the atmosphere and ocean component of the P-SKRIPS model, we aimed at comparing varied air-sea ice drag coefficients for sea ice in the Ross Sea. We varied the standard value of  $1.0 \cdot 10^{-03}$  to  $2.0 \cdot 10^{-03}$  and  $3.0 \cdot 10^{-03}$  and examined the results.

Increasing the drag coefficient leads to a higher mobility of sea ice, which is pushed away from the coast more easily by the winds. The larger area of open water promotes air-sea exchanges and the formation of new sea ice, induced by the strong offshore winds. While the change in drag coefficient does not seem to impact the variability of the sea ice cover greatly, it affects its extent: the sea ice is more easily pushed outside of the box by the wind when using a larger coefficient (Figure 2). This holds true for the decrease of sea ice cover towards the end of the year (austral summer months): simulations 002 and 003 displays a sharper decrease than 001. And because the sea ice is less mobile, the thickness of the snow on the sea ice layer is also larger in simulation 001, albeit marginally (Table S1).

The effect of varying the air-sea ice drag coefficient is small at the atmospheric scale, but the impact varies according to the variable investigated: while the wind and the surface pressure show very minimal changes, the relative humidity and the 2m temperature as well as the latent and sensible heat fluxes are more distinct between the simulations (not shown). There is an increase in both 2 m relative humidity and 2m temperature when using a larger coefficient. Between May and September, the polynyas are active and the sea ice cover reduces with the higher coefficient and the larger polynya generates increased moisture fluxes and a slight warming of the atmosphere, as expected. During these times, the latent and sensible heat fluxes also see an increase between the lower and larger drag coefficient.

The ocean also responds to changes in sea ice formation, as cool and denser saline water is formed due to brine rejection. This is accompanied by heat and humidity loss to the atmosphere. Figure 12 clearly illustrates the variations in ocean salinity arising from sea ice formation in the RSP. Vertical isohalines indicate convection and mixing as the denser water sinks deeper in the ocean when sea ice is created at the surface within the polynya. The larger the drag coefficient, the more saline the water (increase by  $\sim$ 0.2 psu, Figure S8). The same holds true when comparing the model to moorings in the TNBP (Figures 9 to 11), especially at the deeper levels. The model drift over the course of the year in TNBP, present in all simulations, likely indicates problems with the sea ice formation rates in the model in general, which may be improved by coupling with a state-of-the art sea ice model.

# 4.2 Dataset uncertainties and limitations

The spread of the satellite-derived SIC and SIT (Figures 2 and 3) illustrates the complexity of measuring and simulating the sea ice cover. The various algorithms developed to retrieve sea ice extent from satellites are all accompanied by large uncertainties and limitations: e.g. presence of clouds (Spreen et al., 2008), freeboard estimation (Fons et al., 2023)) and surface conditions (Tschudi et al., 2019; Kaleschke et al., 2012), leading to ranges from 10 to 40 %, uncertainty. In addition, coarser satellite products suffer from reduced resolution and mixed ocean-ice cells might be interpreted as ice only. Regarding reanalyses, Nie




et al. (2022) reports that the simulated sea ice cover in ORAs5 and Glorys is realistic, but likely results from compensation errors in thermodynamics, deformation, advection and diffusion of sea ice. In general, the summer season is better represented than the rest of the year.

The grid spacing of the various products and model output explain some of the discrepancies in SIT and SIC, especially along the coast. There, the land-sea mask varies between the datasets that take into account different coastlines (iceberg break up impacts the coastline, for instance) and some cells might be considered fully ocean, fully land or a mix of the two, in the different datasets. The different dataset resolutions and land-sea masks impede the comparison of the same cells along the ice shelf front (Figure S1). Instead, we compare the cells closest to the ice shelf front, in each dataset. This means that we are not comparing cell to cell, but rather location in relation to the ice shelf.

Finally, the difficulties of measuring, integrating observations and parameterizing processes in the ocean is illustrated by the large differences in the spatial patterns and values between the different profiles. The Ross Sea profiles from the WOA result from interpolated points measurements - which explains the smoother transitions and more homogeneous profiles. Because measurements in winter are very scarce, the winter profiles are less reliable. The Glorys and ORAs5 datasets are produced by numerical models that assimilate direct observations from various sources, but these models clearly suffer from limitations. This is especially the case for ORAs5, which has very stratified waters throughout the whole year, and very warm and dense water sitting at the bottom of the ERS, which is not a feature found in the observations. Glorys lies closer to the observations and indicates the presence of convection induced by polynya activity in winter, but is not fresh enough in summer.

# 4.3 P-SKRIPSv2 performances and limitations

The coupled model uses the sea ice component included into the MITgcm model, which was originally designed to help simulate realistic oceanic conditions, but not developed to match and reproduce sea ice physical processes. While the atmospheric and oceanic drivers of sea ice formation and change are well simulated by the coupled model, the response of sea ice might not be realistic, due to the simplifications and limitations of that sea ice routine. However, the current sea ice parametrization is 'good enough' to generate the corresponding response in the ocean through changes in temperature and salinity.

In addition, the coupled model suffers from the lack of measurements of sea ice and ocean conditions, especially under the Ross Ice Shelf, in order to initiate and periodically force the model with realistic values. Regarding the cavity, the workaround is to run a stand-alone simulation for a long period of time, with arbitrary initial conditions and use the state of the cavity at the end of that spin up as initial conditions for the coupled model simulation, with all the limitations this entails.

The coupled model sits in a sweet spot, where its 10 km grid spacing allows it to resolve smaller scale features, but it also fails to compare with point measurements along the coastline or localised mooring data. Its relatively simple sea ice routine is also limiting the comparison to airborne retrievals and does not realistically represent the large heterogeneity of the sea ice cover at small scale.

Despite these, the P-SKRIPS coupled model for the Ross Sea usefully simulates the polynya air-sea ice-ocean system in a realistic manner. The ocean component of the model is one of the few cavity-resolving, an essential feature in the studied area. The spatial resolution of P-SKRIPS is also an advantage to coarser GCMs as its relatively small means it is able to capture

the atmospheric flow in complex terrain, driving the activity of the polynya and resolving smaller scale processes than CMIP6 models. At the regional scale, the fact that P-SKRIPS is a coupled model enables to respond to atmospheric drivers realistically, allowing for feedbacks that are absent in reanalyses. Thus P-SKRIPS spatiotemporal resolution makes it an ideal tool to investigate polynya processes as case studies and for process understanding.

#### 400 5 Conclusions




In this paper, we assess the performance of the coupled P-SKRIPSv2 model for the Ross Sea. We investigate the representation of sea ice and polynyas in the P-SRKIPS model, and assess atmosphere and ocean variables by comparing the simulations to a range of available satellite, in-situ measurements and reanalysis datasets for the year 2017. We also vary the air-sea ice drag coefficient, by doubling and tripling the standard value  $(1.0 \cdot 10^{-3})$  and assess the impact on ocean, sea ice and atmospheric variables.

It comes as no surprise that varying the air-sea ice drag coefficient leads to an increase in polynya activity with more mobile sea ice, sea ice production and brine rejection as well as increased air-sea interactions. While the current sea ice representation is adequate, it could be improved with a dedicated sea ice model that explicitly captures sea ice processes and dynamics. Integrating more than one layer thermodynamics and subgrid scale sea ice thickness distribution would enhance the results and will be the focus of future work.

We conclude that the coupled air-ocean-sea ice model for the Ross Sea is able to simulate realistic polynyas and reproduce the drivers (atmospheric conditions) and the consequences (ocean conditions) during the year 2017 adequately. More specifically, the sea ice cover is underestimated compared to satellite products but matches well with the reanalyses. The modelled sea ice thickness is within the range of two available satellite products, which illustrates the difficulties and limitations of remote sensing retrievals of sea ice in Antarctica. Both the ocean (MITgcm) and the atmosphere (PWRF) perform well and display similar variability and mean values for the main meteorological, sea ice and oceanic variables as observed and reported for the Ross Sea. PWRF performs equally well as AMPS, over both the ocean and land - ice shelf and complex terrain. MITgcm responds to the atmospheric forcing in polynya areas, has mCDW inflow in the Central Ross Sea and is closer to the observations (WOA) than ORA and Glorys in summer.

420 Finally, while recognising the tremendous increase in observation density and availability, we advocate for observational datasets that would fill the spatial and temporal gaps currently observed in the Ross Sea. Specifically, measurements of the water masses within the RIS cavity, as well as better constrained sea ice cover and thickness retrievals would enhance reanalysis products, in turn providing adequate initial and periodic forcing at the boundaries of the simulated domain.

Code and data availability. The P-SKRIPS model code and installation manual can be found at git@github.com:alenamalyarenko/pskrips and the current version (v2) is deposited on zenodo: 10.5281/zenodo.15942874. The model outputs relevant to this manuscript and used to generate the figures are available on Zenodo: 10.5281/zenodo.15778732. The datasets used in this manuscript can be accessed as follows:

430

435

AMSR2 sea ice concentration: https://data.seaice.uni-bremen.de/amsr2/asi

\_daygrid\_swath/s3125/2018/jun/Antarctic3125/; SSMI sea ice concentration: https://www.cen.uni-hamburg.de/en/icdc/data/cryosphere/ seaiceconcentration-asi-ssmi.html; Bootstrap sea ice concentration: https://nsidc.org/data/nsidc-0079/versions/3; CryoSat-2 sea ice concentration and thickness: https://zenodo.org/record/7327711#.ZGWscZFBz0p; SMOS sea ice thickness: https://doi.org/10.1594/PANGAEA.93 4732; NOAA DOISST v2.1 sea ice concentration: https://www.ncei.noaa.gov/data/sea-surface-temperature-optimum-interpolation/v2.1/acces s/avhrr/; Glorys12v1 reanalysis: https://data.marine.copernicus.eu/product/GLOBAL\_MULTIYEAR\_PHY\_001\_030/description; the ORAs5 reanalysis: https://www.ecmwf.int/en/forecasts/dataset/ocean-reanalysis-system-5; the World Ocean Atlas: https://www.ncei.noaa.gov/produc ts/world-ocean-atlas; The AntAWS dataset: https://amrdcdata.ssec.wisc.edu/dataset/antaws-dataset; The PIPERS dataset: https://www.usap-dc.org/view/project/p0010032; The hydrographic DITx moorings: https://kpdc.kopri.re.kr/search/c266365d-4846-4242-952b-75102a53110b; the LDEO moorings: https://academiccommons.columbia.edu/doi/10.7916/a4k3-0a14; and the MORsea moorings can be requested through https://morsea.uniparthenope.it/

Author contributions. AG and AM designed the model set up and ran the experiments. AG and AM analysed the results and produced the figures. LC and UM produced the mooring figures. AM, UM, NL, PC and LC analysed the mooring results. All authors contributed to the writing of the manuscript.

Competing interests. no competing interests are present

Acknowledgements. This research has been supported by the Antarctic Science Platform (grant no. ANTA 1801), and the Antarctic Sea-Ice Switch Programme (contract RSCHTRUSTVIC2448) both funded by the New Zealand Ministry for Business, Innovation and Employment (MBIE). AM is supported by ASP and the University of Canterbury Faculty of Science New Ideas Seeding Grant. The data from moorings B and D were collected as part of the Italian Marine Observatory in the Ross Sea (MORSea) project, which is supported by the Italian National Program for Antarctic Research (PNRA), providing both financial and logistical support.

.

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

| product         | full name                                                                                        | reference                                                           | horizontal resolution                                                             | temporal resolution                                                                           | variables used                                                                                                                                                                                                                           |
|-----------------|--------------------------------------------------------------------------------------------------|---------------------------------------------------------------------|-----------------------------------------------------------------------------------|-----------------------------------------------------------------------------------------------|------------------------------------------------------------------------------------------------------------------------------------------------------------------------------------------------------------------------------------------|
| WOA             | World Ocean Atlas                                                                                | Reagan et al. (2024)                                                | 1/4 degree, 3D                                                                    | monthly                                                                                       | ocean temperature, ocean salt                                                                                                                                                                                                            |
| NOAA            | NOAA OI SST v2.1                                                                                 | Huang et al. (2021)                                                 | 1/4 degree, 2D                                                                    | daily                                                                                         | concentration sea ice cover, sea surface tem-                                                                                                                                                                                            |
| Glorys          | Glorys12v1                                                                                       | Gasparin et al. (2021)                                              | 1/12 degree, 2 and 3 D (50 vertical levels)                                       | daily/monthly                                                                                 | Ocean salinity, ocean temperature, sea ice cover , sea ice thickness                                                                                                                                                                     |
| ORAs5           | Ocean Reanalysis System 5                                                                        | Zuo et al. (2019)                                                   | 1/4 degree, 2 and 3 D (75 vertical levels)                                        | monthly                                                                                       | Ocean salinity, ocean temperature, sea ice cover , sea ice thickness                                                                                                                                                                     |
| Bootstrap       | Bootstrap Sea Ice Concentrations from Nimbus-7 SMMR and DMSP SSM/I-SSMIS, Version 3              | Comiso (2017)                                                       | 25km polar stereo, 2D                                                             | daily                                                                                         | sea ice cover                                                                                                                                                                                                                            |
| SSMI            | Sea Ice Concentrations from Nimbus-7 SMMR and DMSP SSM/I-SSMIS Passive Microwave Data, Version 2 | DiGirolamo et al. (2022)                                            | 12.5km polar stereo, 2D                                                           | daily                                                                                         | sea ice cover                                                                                                                                                                                                                            |
| Cryosat<br>SMOS | CryoSat-2<br>Soil Moisture and Ocean Salinity<br>mission                                         | Fons et al. (2023)<br>Kaleschke et al. (2012)                       | 25km, polar stereo, 2D<br>12.5km, polar stereo, 2D                                | monthly<br>daily                                                                              | Sea ice cover, sea ice thickness<br>sea ice thickness                                                                                                                                                                                    |
| AMSR            | Advanced Microwave Scanning<br>Radiometer 2                                                      | Melsheimer and Spreen (2019)                                        | 3.125km, 2D                                                                       | daily                                                                                         | sea ice cover                                                                                                                                                                                                                            |
| AWS             | Automatic Weather Station                                                                        | Wang et al. (2023)                                                  | point measurement                                                                 | 3-hourly                                                                                      | Atmospheric 2m temperature,<br>2m relative humidity, 10m<br>wind speed, 10m wind direc-<br>tion, surface pressure                                                                                                                        |
| AMPS            | Antarctic Mesoscale Prediction<br>System                                                         | Powers et al. (2003)                                                | 2.67km, 2-3D and 60 vertical levels                                               | hourly                                                                                        | Atmospheric temperature, relative humidity, wind speed, wind direction, surface presents                                                                                                                                                 |
| PIPERS          | Polynyas and Ice Production in the Ross Sea program                                              | Ackley et al. (2020)                                                | vertical profiles at point locations                                              | 11 April to 14 June 2017                                                                      | ocean conductiv- ity-lemperature-depth (CTD) profiles, atmospheric ra- diosondes (temperature and U and V wind components) and atmospheric measure- ment aboard the vessel (air temperature, wind speed and direction, surface pressure) |
| DIT'x<br>L'DEO  | Drygalski Ice Tongue hydrographic<br>mooring time series<br>Lamont-Doherty Earth Observatory     | Cornelissen et al., <i>in prep.</i> Miller et al. (2024a, b); Zappa | 1 long (600m mooring), and 1 short (50m) mooring vertical profiles at point loca- | 9 February to 31 December 2017 1 minute sampling frequency,                                   | Temperature, salinity and depth Temperature, salinity and                                                                                                                                                                                |
| MORSea          | Marine Observatory in the Ross Sea                                                               | and Miller (2018)<br>Zhang et al. (2024)                            | tion<br>vertical profiles at point loca-<br>tion                                  | February 2017 - March 2018<br>30/60 mins sampling frequency, February 2017 -<br>December 2017 | depth<br>Temperature, salinity, depth                                                                                                                                                                                                    |

**Table 2.** Root Means Square Error (RMSE) and Willmott index (WI) for P-SKRIPS sim 002 and AMPS simulations, compared to Vito and Manuela AWS. WI is a standardized way to assess the prediction error of a model, taking into account the systematic and random disparities between the models and the observations, where 1 is a perfect match between the predicted and observed values, and 0 indicates no agreement at all (Willmott et al., 2012)

|                               | RMSE     |       |          |       | Willmott Index |      |          |      |
|-------------------------------|----------|-------|----------|-------|----------------|------|----------|------|
|                               | Manuela  |       | Vito     |       | Manuela        |      | Vito     |      |
|                               | P-SKRIPS | AMPS  | P-SKRIPS | AMPS  | P-SKRIPS       | AMPS | P-SKRIPS | AMPS |
| relative humidity (%)         | 17.69    | 18.99 | 14.22    | 14.16 | 0.53           | 0.36 | 0.40     | 0.42 |
| temperature ( $^{\circ}$ C)   | 3.83     | 3.01  | 6.28     | 4.8   | 0.81           | 0.84 | 0.76     | 0.81 |
| wind speed $(m.s^{-1})$       | 7.11     | 5.65  | 4.14     | 2.99  | 0.65           | 0.73 | 0.41     | 0.53 |
| wind direction ( $^{\circ}$ ) | 36.24    | 43.09 | 60.84    | 60.81 | 0.48           | 0.51 | 0.45     | 0.57 |
| presssure (hPa)               | 15.78    | 12.44 | 5.19     | 2.53  | 0.34           | 0.38 | 0.75     | 0.87 |