# Peer review of "Representation of polynyas in the Ross Sea coupled atmosphere—sea ice—ocean model P-SKRIPSv2"

_EGUsphere, 2025_

## Referee Comment (RC1)

**Representation of polynyas in the Ross Sea coupled atmosphere-sea ice-ocean model P-SKRIPSv2**

**General Comments**

This study uses the coupled P-SKRIPSv2 model in the Ross Sea region and evaluates the model's ability to simulate polynya activity (the Terra Nova Bay Polynya and the Ross Sea Polynya). A sensitivity test on the air—sea ice drag coefficient is also conducted. The results and the model are interesting. However, a major effort to improve readability is needed, including restructuring the manuscript and clarifying the use of datasets and figures. Moreover, I believe that a more thorough analysis of the impact of changes in the drag coefficient on sea ice is needed before the paper is suitable for publication. I also have other scientific questions or comments (on the impact of air-sea drag coefficient on sea ice, the characterization of the polynya activity or the characteristics of the water masses in the region). I have indicated my major comments below.

**Major Comments**

- 1. The presentation of the results is not sufficiently clear, making this section difficult to follow. I recommend splitting it into two distinct parts: one focusing on the representation of polynyas and the various comparisons with observations in each environment, and another dedicated to the sensitivity test results on the air—sea ice drag coefficient, comparing only the outcomes of the sim00x simulations.
- 2. Overall, across both the main article and the supplementary material, it appears that too many datasets are used (including six for sea ice and seven for the ocean—reanalyses and measurements included). Several of these datasets do not seem essential to the study's objectives, as the induced dataset-to-dataset comparisons (outside the scope of the article). Selecting a smaller subset of datasets, or presenting them in a more streamlined manner—for instance by averaging or only showing the minimum or maximum values—would improve readability and help shorten the supplementary material.
- 3. Additionally, this manuscript includes a very large number of figures, some of which could be removed or moved/exchanged to the supplementary material. To give a representative example, the same figure appears twice (Fig. 8 and Fig. S7). Panel numbers should be added to the figures and referred to in the text. The colorbars are not always very legible, and the plot colors are often difficult to distinguish. In some cases, labels or colorbars are missing.
- 4. It is not clear from this analysis how, and to what extent, changes in the air-ice drag coefficient affect the sea ice field (e.g. with maps):
  - a. How are ice thickness and concentration affected in the region?
  - b. What are the effects on ice drift and ice production within the polynyas?
- 5. This paper focuses on the representation of polynyas, but no quantities related to polynya activity are presented. For example:
- a. What are the areas, shapes, and sea-ice production of the polynyas in the region compared to observations (e.g., Nakata et al., 2021: <a href="https://doi.org/10.1029/2020GL091353">https://doi.org/10.1029/2020GL091353</a>)?
  - b. How do these quantities change in response to variations in the air-ice drag coefficient?
- 6. Looking at the mooring measurements (Figs. 9, 10, 11), the simulated salinities appear to differ significantly from the observations. Could you present the characteristics of the water masses simulated by the model compared to observations (e.g., using T–S diagrams)?

**Specific Comments**

Please add numbers or letters to each subfigure (as in Fig.1)

**Abstract**

• You could be a bit more explicit about your results (e.g., regarding the model's performance or the impact of the air–sea ice drag coefficient).

**Introduction**

- L21: You may also consider adding the following references: Golledge et al. (2025) (<a href="https://doi.org/10.1038/s43017-024-00634-x">https://doi.org/10.1038/s43017-024-00634-x</a>) and Noel et al. (2025) (<a href="https://doi.org/10.1029/2025JD043319">https://doi.org/10.1029/2025JD043319</a>), which respectively review and address this question in the context of Antarctic coastal polynyas.
- L30-39: This paragraph is confusing for the rest of the study. The ocean-ice parameter seems to be introduced and mixed with the air-ice drag parameter together, even though the sensitivity test focuses solely on the air-ice drag coefficient.
- L44: Replace (( with (
- L45: You may also consider mentioning the difficulty that CMIP6 GCMs have in forming dense waters such as AABW (Heuzé, 2021 <a href="https://doi.org/10.5194/os-17-59-2021">https://doi.org/10.5194/os-17-59-2021</a>).
- L55: The term "response to the polynya extent" is somewhat misleading, as the polynyas' area (or their evolution) is not provided elsewhere in the study.

**Methods**

- L66-67: I did not understand what you meant.
- In subsection 2.1, I would have appreciated if you specified which forcings were used, how
  the model was initialized, and whether a spin-up was applied for the ocean—and if so, for how
  long.
- L83: It is stated that the default simulation is sim001, but in several figures, sim002 is used as the reference (e.g., Fig. 5, Fig. 6, Fig. 8). Why is this?
- L84-86: The formulation is unclear.
- Subsection 2.3 should be revised or better introduced (see Major Comments above).
- Note that, to improve readability throughout the manuscript, you could consider referring to each of the moorings not by their full names, but simply as "moorings" + number/letter or by using an acronym or a short descriptive label and which could be added to the legend of Figure 1c.
- L103: Change "start" to "stars" and indicate in Figure 1c which ones are DITN and DITD.

**Results**

I will remain brief on this part, which should be restructured and partially rewritten by the authors (see Major Comments above).

- In subsection 3.1 on Sea Ice, too many products are discussed, which dilutes the focus on the simulation itself. In addition, the time series alone are insufficient; mean maps of ice thickness or concentration are missing, which are needed to properly discuss the model's representation of sea ice and would allow the characterization of polynyas in the simulations and observations. Including these would strengthen the results (L142–146), especially regarding the differences caused by the boundary conditions (L145). The subsection on sea-ice thickness should be shortened. The colors of the plots in Figures 2 and 3 should be changed, as the curves are hardly distinguishable.
- Subsection 3.2 is particularly interesting. In Figure 4, the x-axis should be changed to show the month names, as in the other figures (e.g., Fig. 2 and 3), and the pink color is difficult to read.
- L213-214: The terms "fairly close" and "quasi-overlap" should be qualified when comparing the distributions of relative humidity, wind speed, and pressure (Fig. 6 and Fig. S5).
- L216: For "largest RMSE," please provide the actual values.
- In subsection 3.4, the colorbar used in Figure 7 is not clear. Similarly, Figure 8 should be revised: the maximum depth needs to be changed, there is no legend for the top panels, the bottom panels are hard to read, and the colors are illegible.
- Sections 3.5 and 3.6 are difficult to follow, and the figures are hard to read (Figs. 9, 10, 11, 12, 13), with poorly legible colorbars and colors. Additionally, in section 3.5, the errors in salinity of the simulations compared to observations (Figs. 9, 10, 11) are barely discussed. Perhaps,

before discussing salinity/temperature variability, the water masses in the region could be characterized (e.g., TS diagrams) in the simulations compared to one of the ocean products/ reanalyses.

**Discussion**

The discussion is interesting, but it seems to me that subsection 4.2, dealing with the uncertainties of the datasets, falls outside the scope of the article.

- L339-340: Provide values for the simulations.
- L344: Values ?L347: Values ?

**Conclusions**

Very good conclusion

---

## Referee Comment (RC2)

Review of "Representation of polynyas in the Ross Sea coupled atmosphere-sea ice-ocean model P-SKRIPSv2" by Gossart et al.

This review is co-signed by François Massonnet and Noé Pirlet (UCLouvain)

**Summary**

In this study, Gossart et al. present P-SKRIPSv2, a regional atmosphere-ocean-sea ice model based on the WRF atmospheric model, the MITgcm ocean model and the Semtner 0-layer sea ice model with viscous plastic rheology, to understand the importance of coastal polynya formation processes in the Ross Sea and the sensitivity of the model results to the choice of the air-sea ice drag coefficient.

**General comments**

- The paper does a good job at comparing model output with a wealth of observational data (from satellite, reanalyses, cruises, hydrographic moorings, …) but is at times lengthy and quite descriptive.
- We have the impression that also the scientific question of the paper is not entirely clear to us. Part of the paper is used to evaluate the model while part of the paper is about sensitivity tests on the drag coefficient. Since the science question is not clear, the choice of the domain is not entirely clear either.
- The paper would be more suited to GMD in the present form, since it is essentially a model evaluation study and not a study about a physical process
- The structure of the manuscript should be consistent across the title, abstract, results, discussion, and conclusion. We recommend clearly defining the central research question and reorganizing the manuscript around it.
- In evaluating the realism of the simulated polynyas, the authors do not compare their extent, shapes, or the associated sea-ice production with existing observational estimates (e.g., Nihashi and Ohshima, 2015; Nakata et al., 2021) or model estimations (e.g. Pirlet et al., 2025). If the goal of the paper (and the central scientific question) is about polynya formation, then adequate diagnostics should be used.

**Positioning with respect to the state-of-the-art**

- The very recently published paper of (Pirlet et al., 2025) is a key one to cite. We understand that the authors may have not seen it when it was published, but it would be good to position the current paper with respect to this paper (about modeling of Antarctic coastal polynyas). We also encourage the authors to read the (less recent) papers by (Huot et al., 2021) and (Van Achter et al., 2022) where similar questions are treated.
- The paper of Pelletier et al., (2022) might also be worth having a look at since it covers similar aspects (fully coupled model) to what is encountered with P-SKRIPSv2.
- The paper Noel et al., (2025) about coastal polynya-atmosphere feedback is also missing in the introduction.

**Methodological questions**

- Line 70: does P-SKRIPSv2 account for snow-ice formation, which can be quite an important process in the Southern Ocean?
- Even though the study is largely based on the previously published paper of Malyarenko et al. 2023, it would be useful to repeat (1) what the boundary conditions (ocean and atmosphere) are or the model setup (especially since these boundary conditions are said to be a cause of model error, see lines 145,280,316), (2) what was the tuning procedure for the model, (3) whether the model was spun-up or not.
- The methods are a bit shallow regarding the period used. From the figures we deduce that 2017 was chosen, but then several questions come up: why choosing only one year to perform a model evaluation; and why choosing that year in particular?
- In Fig. 2, the "polynya" regime is defined as when SIC is equal to 0.6-0.8. Where does that number come from? SIC can take, on average over the domain, the same value for many possible configurations, including ones that do not have polynyas. For model data, SIT is (additionally) employed to detect coastal polynyas, as it helps mitigate the model's tendency to overproduce sea ice and thereby prevents polynyas exhibiting near-100% concentrations for unrealistic reasons from being missed. Could another threshold or variable change your results? Can you motivate your choice ?
- How about tides, waves, ice shelves? Are these processes relevant for polynya formation, and if yes, are they captured / accounted for by the model? If not, what are the implications on the results, on the realism of the simulation?
-  The area studied in this paper is infested with icebergs, which have huge impact on the landfast ice and then polynyas. Icebergs can also modify water masses when they release freshwater. Does that affect the model results?
- The Willmott index used in Table 2 is a rather unusual one, consider detailing its meaning in the section on methods. Also, this acronym is introduced before being cited.
- Subsection 2.3 is quite dense as a single paragraph; we recommend splitting it into several shorter paragraphs, for instance one for each type of dataset.

**Other remarks**

- The abstract is very short and does not render the breadth of the findings of this study. The authors should consider including more context, more results, and more perspectives.

**Writing**

The writing is in general impeccable here is the few typos we found

- L44: )) -> )
- L103: )) -> ) to times
- L104: )) -> )
- L103: start -> stars (yellow) and stars -> star (purple)
- L106: )) -> )
- L113: )) -> )
- L154: "larger" than what ?

- L189: man -> many?
- L191: need a space between "Figure 5" and "as" otherwise it's confusing
- L200: W.m^-2 is missing, also a ";" is missing after "heat"; on that line, use the proper symbol for +/-
- L202: remove "and" after "wind"
- L222: )) -> )
- L339: displays -> display
- L359: )) -> )

Suggestions:

- L6: We propose "coastal polynyas activities" instead of "polynya activity".
- L49: You could directly mention here which are the coastal polynyas instead of later as in L52.
- L158: We do not understand the sentence "Satellite products also show insufficient variation over the year.".
- L166: "AR" is never defined.
- L217: the acronym "WI" is used but only introduced in a table later; consider expanding it here too.
- L411: Maybe use "realistic coastal polynyas activities" with "realistic polynyas" to be coherent with the rest of the paper.

**Figures**

- Figure 1:
  - Change "simulated domain" to "domain of simulation" since a domain is not simulated.
  - In panel c, consider adding the text of the different items (red box, red star, green start, etc.) directly in the figures near the symbols, this would be much more visual.
- Figure 2:
  - We would propose to change the color of the lines to make the distinction between satellite, reanalysis, and model output. Cryosat, AMSR, NOAA, SSMI and Bootstrap would go in shades of blue; GLORYS and ORA would go in shades of red; and the three sims would go in shades of green (for example). The use of dash vs solid lines is a bit confusing.
  - Alternatively, consider using a shading to display the range of observations.
- Figure 5: There is a "a)" label, but no figure panel corresponds to it.
- Figure 6:
  - We suggest merging it with Figure 5, as both figures share the same data.
  - If we get it right, in the last sentence: the AMPS dataset should be indicated in blue (not pink), and conversely the AWS data should be shown in pink (not blue).
- Figure 7-8: Axis labels are not always necessary on every panel; we suggest keeping them only on the outer edges of the figures. This recommendation could also apply to other figures.
- Figure 9-10-11: We suggest merging these figures, as they provide similar information, and moving some of the panels to the Supplementary Material.

- Figure 12-13:
    - We are not clear why the bathymetry of the model and the references are so different. Can the authors clarify this?
    - The legend is not sufficiently detailed or explicit.

**References**

Huot, P.-V., Fichefet, T., Jourdain, N. C., Mathiot, P., Rousset, C., Kittel, C., and Fettweis, X.: Influence of ocean tides and ice shelves on ocean–ice interactions and dense shelf water formation in the D'Urville Sea, Antarctica, Ocean Model., 162, 101794, https://doi.org/10.1016/j.ocemod.2021.101794, 2021.

Nakata, K., Ohshima, K. I., and Nihashi, S.: Mapping of Active Frazil for Antarctic Coastal Polynyas, With an Estimation of Sea‑Ice Production, Geophys. Res. Lett., 48, e2020GL091353, https://doi.org/10.1029/2020GL091353, 2021.

Nihashi, S. and Ohshima, K. I.: Circumpolar Mapping of Antarctic Coastal Polynyas and Landfast Sea Ice: Relationship and Variability, J. Clim., 28, 3650–3670, https://doi.org/10.1175/JCLI-D-14-00369.1, 2015.

Noel, M., Masson, S., and Rousset, C.: Atmospheric response to Antarctic coastal polynyas, , https://doi.org/10.5194/egusphere-egu25-2877, 2025.

Pelletier, C., Fichefet, T., Goosse, H., Haubner, K., Helsen, S., Huot, P.-V., Kittel, C., Klein, F., Le Clec'H, S., Van Lipzig, N. P. M., Marchi, S., Massonnet, F., Mathiot, P., Moravveji, E., Moreno-Chamarro, E., Ortega, P., Pattyn, F., Souverijns, N., Van Achter, G., Vanden Broucke, S., Vanhulle, A., Verfaillie, D., and Zipf, L.: PARASO, a circum-Antarctic fully coupled ice-sheet–ocean–sea-ice–atmosphere–land model involving f.ETISh1.7, NEMO3.6, LIM3.6, COSMO5.0 and CLM4.5, Geosci. Model Dev., 15, 553–594, https://doi.org/10.5194/gmd-15-553-2022, 2022.

Pirlet, N., Fichefet, T., Vancoppenolle, M., Fraser, A. D., Mathiot, P., Rousset, C., Barthélemy, A., Barriat, P. -Y., Pelletier, C., Madec, G., and Kittel, C.: Benefits of a Landfast Ice Representation on Simulated Antarctic Sea Ice and Coastal Polynya Dynamics, J. Geophys. Res. Oceans, 130, e2024JC022032, https://doi.org/10.1029/2024JC022032, 2025.

Van Achter, G., Fichefet, T., Goosse, H., Pelletier, C., Sterlin, J., Huot, P.-V., Lemieux, J.-F., Fraser, A. D., Haubner, K., and Porter-Smith, R.: Modelling landfast sea ice and its influence on ocean–ice interactions in the area of the Totten Glacier, East Antarctica, Ocean Model., 169, 101920, https://doi.org/10.1016/j.ocemod.2021.101920, 2022.

---

## Author Comment (AC1)

Representation of polynyas in the Ross Sea coupled atmosphere-sea ice-ocean model P-SKRIPS
by: Gossart, A., Malyarenko, A., Cornelissen, L. and Stevens, C.

December 2025

We are very grateful to the three reviewers for their thorough review of our paper. Their remarks and suggestions are very valuable and have led us to reconsider the scope of the manuscript.

Before the detailed response to the comments, we wish to respond to major concerns raised by both reviewers regarding the structure, focus and analysis present in the preprint. Both reviewers raised the issue of the length and density of the manuscript, and that the drag coefficient experiments were not fully exploited. Therefore, we decided to remove the analysis pertaining to the variation in drag coefficient from the present manuscript and are instead concentrating on the simulation using the default coefficient only. We will compare the simulation results to available observations and the coefficient study will be the object of another manuscript. Furthermore, we will expand the sea ice analysis by adding seasonal maps of ice cover and thickness, as well as computing estimates of sea ice production. With this new focus in mind, and to avoid creating confusion, we propose to change the title of the manuscript to "Representation of the Ross Sea Region coupled atmosphere-sea ice-ocean system using P-SKRIPSv2"". We have reduced the number of datasets to which we compare our results- and reduced the number of figures, which is reflected in the change in authorship. We hope this new version of the paper is more streamlined, clear and easier to read.

Below are the detailed response to the reviewer comments, including comments and future actions by the authors:

**Reviewer 1**

**Major Comments**

1. The presentation of the results is not sufficiently clear, making this section difficult to follow. I recommend splitting it into two distinct parts: one focusing on the representation of polynyas and the various comparisons with observations in each environment, and another dedicated to the sensitivity test results on the air–sea ice drag coefficient, comparing only the outcomes of the sim00x simulations.

We apologise that the results section is lengthy and hard to follow. As explained above, we will leave out the sensitivity study in the revised version of the manuscript. We will also limit the comparison to the satellite and reanalysis estimates of the sea ice (cover and thickness, figures 2 and 3). The performance of the atmospheric model will be assessed in Terra Nova Bay (figure 4) and using distribution plots at the Manuela weather station (figure6). The oceanic results will be compared to the DITx moorings (figure 9), and the temperature and salinity profiles comparison will be limited to the Glorys reanalysis only. Therefore, figures 5, 7 and 8, 10 and 11 will be removed, and figures 12 and 13 will be reduced and merged. Figures S6 to S9 will be removed, and figures S10 and 11 will be merged with figures 12 and 13.

2. Overall, across both the main article and the supplementary material, it appears that too many datasets are used (including six for sea ice and seven for the ocean—reanalyses and measurements included). Several of these datasets do not seem essential to the study's objectives, as the induced dataset-to-dataset comparisons (outside the scope of the article). Selecting a smaller subset of datasets, or presenting them in a more streamlined manner—for instance by averaging or only showing the minimum or maximum values—would improve readability and help shorten the supplementary material.

*As stated above, we will reduce the amount of datasets and streamline the writing of the results.*

3. Additionally, this manuscript includes a very large number of figures, some of which could be removed or moved/exchanged to the supplementary material. To give a representative example, the same figure appears twice (Fig. 8 and Fig. S7). Panel numbers should be added to the figures and referred to in the text. The colorbars are not always very legible, and the plot colors are often difficult to distinguish. In some cases, labels or colorbars are missing.
*We have removed a series of figures, and figures 8 and S7 will not be part of the new version of the manuscript. Moreover, we will ensure that all figures have legible labels and legends, colorbars, and panel numbers.*

4. It is not clear from this analysis how, and to what extent, changes in the air–ice drag coefficient affect the sea ice field (e.g. with maps): a. How are ice thickness and concentration affected in the region?
b. What are the effects on ice drift and ice production within the polynyas?
*Both reviewers raised the issue of the length and density of the manuscript, and that the drag coefficient experiments were not fully exploited. Therefore, we decided to remove the analysis pertaining to the variation in drag coefficient from the present manuscript and are instead concentrating on the simulation using the default coefficient only. We will also include seasonal maps of sea ice cover and thickness as well as an estimate of sea ice production in the Ross Sea region.*

5. This paper focuses on the representation of polynyas, but no quantities related to polynya activity are presented. For example:
a. What are the areas, shapes, and sea-ice production of the polynyas in the region compared to observations (e.g., Nakata et al., 2021: https://doi.org/10.1029/2020GL091353)?
b. How do these quantities change in response to variations in the air–ice drag coefficient?
*a) We agree with this comment and propose to change the title of the manuscript to "Representation of the Ross Sea Region coupled atmosphere-sea ice-ocean system using P-SKRIPSv2"". Because the seaice routine in MITgcm does not simulate the physics of ice as a standalone sea ice model would, we had intentionally left the sea ice production and spatial analysis of sea ice cover and thickness out of the manuscript. We see now that it has its place in the paper and will expand the section accordingly.*
*b) we will leave out the sensitivity analysis in the updated version of the manuscript.*

6. Looking at the mooring measurements (Figs. 9, 10, 11), the simulated salinities appear to differ significantly from the observations. Could you present the characteristics of the water masses simulated by the model compared to observations (e.g., using T–S diagrams)?
*We will remove Figures 10 and 11 from the new version of the manuscript. We will also discuss the water masses present in Figure 9 with the support of a T-S diagram*

**Specific Comments**

Please add numbers or letters to each subfigure (as in Fig.1)
*We will add the letters to each subpanel of each figure.*

**Abstract**
You could be a bit more explicit about your results (e.g., regarding the model's performance or the impact of the air–sea ice drag coefficient).
*We will remove the sensitivity analysis from the study and will expand the abstract to present the results more explicitly.*

**Introduction**

L21: You may also consider adding the following references: Golledge et al. (2025) (https://doi.org/10.1038/s43017-024-00634-x) and Noel et al. (2025) (https://doi.org/ 10.1029/2025JD043319), which respectively review and address this question in the context of Antarctic coastal polynyas.

We thank you for pointing us to these references, we will add them to the manuscript.

L30-39: This paragraph is confusing for the rest of the study. The ocean–ice parameter seems to be introduced and mixed with the air-ice drag parameter together, even though the sensitivity test focuses solely on the air–ice drag coefficient.

We will remove the sensitivity analysis from the study, it will be the focus of another manuscript.

L44: Replace (( with (

We will do it.

L45: You may also consider mentioning the difficulty that CMIP6 GCMs have in forming dense waters such as AABW (Heuzé, 2021 — https://doi.org/10.5194/os-17-59-2021).

We will add this comment and add the reference.

L55: The term "response to the polynya extent" is somewhat misleading, as the polynyas' area (or their evolution) is not provided elsewhere in the study.

We will expand the sea ice analysis and adapt the text accordingly.

**Methods**

L66-67: I did not understand what you meant.

This sentence explains that the sea ice routine is not very elaborate in MITgcm. The sea ice component was designed to help simulate realistic oceanic conditions, but not developed to reproduce sea ice physical processes. However, it is "good enough" to be used in the current coupled model set-up that focuses on conserving the heat and mass fluxes between the ocean and atmosphere models. We will rephrase this sentence and add a reference to MITgcm documentation (https://mitgcm.readthedocs.io/en/latest/phys_pkgs/seaice.html#thermodynamics and https://mitgcm.readthedocs.io/en/latest/phys_pkgs/seaice.html#compatibility-with-ice-thermodynamics-package-pkg-thsice).

In subsection 2.1, I would have appreciated if you specified which forcings were used, how the model was initialized, and whether a spin-up was applied for the ocean—and if so, for how long.

We apologise for this oversight, the text was lost between manuscript versions. We will add a description of the spin-up performed prior to coupled run: The MITgcm ocean model is run in stand-alone mode for the period 1992-2017, since none of the available reanalysis datasets possess a cavity under the ice shelf. MITgcm needs a relatively long spinup to flush out the cavity and create stable ocean conditions in the area. The physical setup for this model is identical to Malyarenko et al. (2023). The initial conditions come from the World Ocean Atlas 2018 (WOA; Locarnini et al. (2018)), interpolated to extend into the cavity. The model bathymetry, including cavity shape, is based on Bedmap2 dataset (Fretwell et al. 2013). The boundary conditions for ocean and sea ice variables at the edge of the domain are taken from monthly mean fields of ECCO Latitude-Longitude-polar Cap 270 (ECCO-LLC270; Zhang, Menemenlis, and Fenty (2018)); tidal forcing is applied at lateral boundaries from CATS2008 model (an update to the model described by Padman et al. (2002)). The atmosphere fields come from ERA-5 reanalysis (Hersbach et al. n.d.). Downward shortwave and longwave radiation, precipitation, humidity and air temperature are provided 6-hourly, while the U and V components of wind are provided hourly. The sea ice - atmosphere drag coefficient was set to $2.10^{-3}$.

The coupled model will also be described as follows: The run starts from the 1st of February 2017, when the sea ice extent is at its minimum and ends in December 2017, allowing a full sea ice yearly cycle. The year 2017 was chosen due to the availability of ocean observations (DITx moorings in Terra Nova Bay area). The

ocean model initial conditions are extracted from the spin up run for the first of February 2017. The monthly forcing of the model by ocean boundary conditions also comes from ECCO-LLC270. The atmosphere model is forced by ERA-5 reanalysis every 6 hours and the physics options and model set up are identical to Malyarenko et al. (2023). We replicate the sea ice concentration from the ERA5 reanalysis for the 1st of February, in order to conserve continuity with the atmosphere, but this field subsequently evolves freely within the ocean model during the simulation. Sea ice information at the boundaries of the domain are supplied by ECCO llc270 (concentration, U and V velocities). Since little information is available on sea ice thickness and snow on sea ice thickness **kaleschke˙tian˙2023**; Kern, Ozsoy-Çiçek, and Worby (2016) and Fons, Kurtz, and Bagnardi (2023) , we initiate our model with a spatially uniform value of 0.5 m sea ice thickness and 0.05 m snow on sea ice thickness for every grid cell covered by sea ice.

L83: It is stated that the default simulation is sim001, but in several figures, sim002 is used as the reference (e.g., Fig. 5, Fig. 6, Fig. 8). Why is this?
We will remove simulations 002 and 003 from the analysis.

L84-86: The formulation is unclear.
This text will be removed in the new version of the manuscript.

Subsection 2.3 should be revised or better introduced (see Major Comments above).
We will revise the section and streamline the writing.

Note that, to improve readability throughout the manuscript, you could consider referring to each of the moorings not by their full names, but simply as "moorings" + number/letter or by using an acronym or a short descriptive label and which could be added to the legend of figure 1c.
We will only keep the DITx moorings in the new version of the manuscript, and will add labels to figure 1 c).

L103: Change "start" to "stars" and indicate in figure 1c which ones are DITN and DITD.
We will adapt the text and update figure 1.

**Results**
I will remain brief on this part, which should be restructured and partially rewritten by the authors (see Major Comments above).
In subsection 3.1 on Sea Ice, too many products are discussed, which dilutes the focus on the simulation itself. In addition, the time series alone are insufficient; mean maps of ice thickness or concentration are missing, which are needed to properly discuss the model's representation of sea ice and would allow the characterization of polynyas in the simulations and observations. Including these would strengthen the results (L142–146), especially regarding the differences caused by the boundary conditions (L145). The subsection on sea-ice thickness should be shortened. The colors of the plots in figures 2 and 3 should be changed, as the curves are hardly distinguishable.
We will rewrite this section and will strengthen the focus. As described in the response to Major comments, we will add maps of sea ice concentration and thickness that will strengthen the discussion. We will also adapt the font and colors of figures 2 and 3.

Subsection 3.2 is particularly interesting. In figure 4, the x-axis should be changed to show the month names, as in the other figures (e.g., Fig. 2 and 3), and the pink color is difficult to read.
We are pleased that the section is interesting to the reviewer. We will update figure 4 accordingly.

L213-214: The terms "fairly close" and "quasi-overlap" should be qualified when comparing the distributions of relative humidity, wind speed, and pressure (Fig. 6 and Fig. S5).

*We will make these changes.*

L216: For "largest RMSE," please provide the actual values.
*We will provide the actual values and variable names.*

In subsection 3.4, the colorbar used in figure 7 is not clear. Similarly, figure 8 should be revised: the maximum depth needs to be changed, there is no legend for the top panels, the bottom panels are hard to read, and the colors are illegible.
*Figures 7 and 8 will be removed from the new version of the manuscript.*

Sections 3.5 and 3.6 are difficult to follow, and the figures are hard to read (Figs. 9, 10, 11, 12, 13), with poorly legible colorbars and colors. Additionally, in section 3.5, the errors in salinity of the simulations compared to observations (Figs. 9, 10, 11) are barely discussed. Perhaps, before discussing salinity/temperature variability, the water masses in the region could be characterized (e.g., TS diagrams) in the simulations compared to one of the ocean products/ reanalyses.
*We will reduce section 3.5 to the DITx moorings and improve the readability of the text. We will increase the readability of figure 9 and remove figures 10 and 11. We will assess the water masses in the continental shelf of the Ross Sea by comparing TS-diagrams to the GLORYS dataset, and change current figure 9 (timeseries) to a TS-diagram as well, as it will explain the differences in a clearer way. We will improve the readability of figures 12 and 13 that will be merged together.*

**Discussion**
The discussion is interesting, but it seems to me that subsection 4.2, dealing with the uncertainties of the datasets, falls outside the scope of the article.
*We disagree and think the uncertainties and variability in observations are useful to understand the model results. Since we now limit the scope of the manuscript to a model-observations comparison (removing the sensitivity study), we feel this section has its place in the manuscript.*

L339-340: Provide values for the simulations.
*Section 4.1 will be removed from the new version of the manuscript.*

L344: Values ?
*Section 4.1 will be removed from the new version of the manuscript.*

L347: Values ?
*Section 4.1 will be removed from the new version of the manuscript.*

**Conclusions**
Very good conclusion.
*Thank you!*

**Review of "Representation of polynyas in the Ross Sea coupled atmosphere-sea ice-ocean model P-SKRIPSv2" by Gossart et al. This review is co-signed by François Massonnet and Noé Pirlet (UCLouvain)**

**Summary**

In this study, Gossart et al. present P-SKRIPSv2, a regional atmosphere-ocean-sea ice model based on the WRF atmospheric model, the MITgcm ocean model and the Semtner 0-layer sea ice model with viscous plastic rheology, to understand the importance of coastal polynya formation processes in the Ross Sea and the sensitivity of the model results to the choice of the air-sea ice drag coefficient.

**General comments**

The paper does a good job at comparing model output with a wealth of observational data (from satellite, reanalyses, cruises, hydrographic moorings, ...) but is at times lengthy and quite descriptive.
We thank the reviewers for recognizing our work. As stated above, we will re-arrange the paper, remove the sensitivity analysis and some figures and strengthen the focus of the paper by including sea ice cover and thickness maps, as well as estimate sea ice production in the Ross Sea.

We have the impression that also the scientific question of the paper is not entirely clear to us. Part of the paper is used to evaluate the model while part of the paper is about sensitivity tests on the drag coefficient. Since the science question is not clear, the choice of the domain is not entirely clear either.
We apologise for the confusion. We refer the reviewers to the general response to reviewers at the top of this document for a precise description of the revised scope of the new version of the manuscript. We also propose to change the title of the manuscript to "Representation of the Ross Sea Region coupled atmosphere-sea ice-ocean system using P-SKRIPSv2"".

The paper would be more suited to GMD in the present form, since it is essentially a model evaluation study and not a study about a physical process.
The first P-SKRIPS paper (coupling work, Malyarenko et al. (2023)) was published in GMD. Since we remove the coefficient sensitivity study from this mansucript, we feel there is no susbstantial model development to qualify for GMD. The focus of the new version of this manuscript will essentially be the validation of the representation of the state of the Ross Sea Region in the coupled model, by comparing to available air, sea ice and ocean observations.

The structure of the manuscript should be consistent across the title, abstract, results, discussion, and conclusion. We recommend clearly defining the central research question and reorganizing the manuscript around it.
We thank the reviewers for this observation. Removing the sensitivity study will make "how does the model result compare to "reality/the observations" the main goal of the manuscript and additional sea ice analysis will hopefully fill the current gap.

In evaluating the realism of the simulated polynyas, the authors do not compare their extent, shapes, or the associated sea-ice production with existing observational estimates (e.g., Nihashi and Ohshima, 2015; Nakata et al., 2021) or model estimations (e.g. Pirlet et al., 2025). If the goal of the paper (and the central scientific question) is about polynya formation, then adequate diagnostics should be used.
In the new version of the new manuscript, we intent to distance from polynya formation and extent, but also to add the sea ice quantities suggested by the reviewers.

**Positioning with respect to the state-of-the-art**

The very recently published paper of (Pirlet et al., 2025) is a key one to cite. We understand that the authors

may have not seen it when it was published, but it would be good to position the current paper with respect to this paper (about modeling of Antarctic coastal polynyas). We also encourage the authors to read the (less recent) papers by (Huot et al., 2021) and (Van Achter et al., 2022) where similar questions are treated. The paper of Pelletier et al., (2022) might also be worth having a look at since it covers similar aspects (fully coupled model) to what is encountered with P-SKRIPSv2. The paper Noel et al., (2025) about coastal polynya-atmosphere feedback is also missing in the introduction.

We thank the reviewers for bringing these papers to our attention. Indeed, the recently published paper by Pirlet et al., as well as several others came to our attention only after submission of the present manuscript. We will make sure to add references to these papers in the new version of the manuscript.

**Methodological questions**

Line 70: does P-SKRIPSv2 account for snow-ice formation, which can be quite an important process in the Southern Ocean?

Yes, the sea ice routine in MITgcm accounts for flooding. We will add this to the text.

Even though the study is largely based on the previously published paper of Malyarenko et al. 2023, it would be useful to repeat (1) what the boundary conditions (ocean and atmosphere) are or the model setup (especially since these boundary conditions are said to be a cause of model error, see lines 145,280,316), (2) what was the tuning procedure for the model, (3) whether the model was spun-up or not.

We apologise for this oversight, the text was lost between manuscript versions. We will add a description of the spin-up performed prior to coupled run: The MITgcm ocean model is run in stand-alone mode for the period 1992-2017, since none of the available reanalysis datasets possess a cavity under the ice shelf. MITgcm needs a relatively long spinup to flush out the cavity and create stable ocean conditions in the area. The physical setup for this model is identical to Malyarenko et al. (2023). The initial conditions come from the World Ocean Atlas 2018 (WOA; Locarnini et al. (2018)), interpolated to extend into the cavity. The model bathymetry, including cavity shape, is based on Bedmap2 dataset (Fretwell et al. 2013). The boundary conditions for ocean and sea ice variables at the edge of the domain are taken from monthly mean fields of ECCO Latitude-Longitude-polar Cap 270 (ECCO-LLC270; Zhang, Menemenlis, and Fenty (2018)); tidal forcing is applied at lateral boundaries from CATS2008 model (an update to the model described by Padman et al. (2002)). The atmosphere fields come from ERA-5 reanalysis (Hersbach et al. n.d.). Downward shortwave and longwave radiation, precipitation, humidity and air temperature are provided 6-hourly, while the U and V components of wind are provided hourly. The sea ice - atmosphere drag coefficient was set to $2.10^{-3}$.

The coupled model will also be described as follows: The run starts from the 1st of February 2017, when the sea ice extent is at its minimum and ends in December 2017, allowing a full sea ice yearly cycle. The year 2017 was chosen due to the availability of ocean observations (DITx moorings in Terra Nova Bay area). The ocean model initial conditions are extracted from the spin up run for the first of February 2017. The monthly forcing of the model by ocean boundary conditions also comes from ECCO-LLC270. The atmosphere model is forced by ERA-5 reanalysis every 6 hours and the physics options and model set up are identical to Malyarenko et al. (2023). We replicate the sea ice concentration from the ERA5 reanalysis for the 1st of February, in order to conserve continuity with the atmosphere, but this field subsequently evolves freely within the ocean model during the simulation. Sea ice information at the boundaries of the domain are supplied by ECCO llc270 (concentration, U and V velocities). Since little information is available on sea ice thickness and snow on sea ice thickness **kaleschke˙tian˙2023**; Kern, Ozsoy-Çiçek, and Worby (2016) and Fons, Kurtz, and Bagnardi (2023) , we initiate our model with a spatially uniform value of 0.5 m sea ice thickness and 0.05 m snow on sea ice thickness for every grid cell covered by sea ice.

The methods are a bit shallow regarding the period used. From the figures we deduce that 2017 was chosen, but then several questions come up: why choosing only one year to perform a model evaluation; and why

choosing that year in particular?

We were limited to years prior to 2017 due to the availability of ECCO llc270, and chose to simulate 2017 because of the observations availability (PIPERS cruise, now removed from the manuscript - and mooring datasets). We will make sure to detail the choice in the new version of the manuscript.

In Fig. 2, the "polynya" regime is defined as when SIC is equal to 0.6-0.8. Where does that number come from? SIC can take, on average over the domain, the same value for many possible configurations, including ones that do not have polynyas. For model data, SIT is (additionally) employed to detect coastal polynyas, as it helps mitigate the model's tendency to overproduce sea ice and thereby prevents polynyas exhibiting near-100% concentrations for unrealistic reasons from being missed. Could another threshold or variable change your results? Can you motivate your choice ?

We thank the reviewers for this comment. We have decided to focus on SIC and SIT, temporal and spatial extent, as well as sea ice production, but will remove the references to polynyas and polynya thresholds from the new version of the manuscript.

How about tides, waves, ice shelves? Are these processes relevant for polynya formation, and if yes, are they captured / accounted for by the model? If not, what are the implications on the results, on the realism of the simulation?

Tides and ice shelves are present in P-SKRIPS, we will add these elements to the model description. Waves are not part of the current coupled model set up. As stated above, we will not mention polynya formation in the new version of the manuscript.

The area studied in this paper is infested with icebergs, which have huge impact on the landfast ice and then polynyas. Icebergs can also modify water masses when they release freshwater. Does that affect the model results?

We observe the presence of landfast ice in McMurdo Sound mostly, which is not very well represented in the model and not discussed in the paper. Icebergs and landfast ice is tricky to represent, especially at 10 km spatial resolution, and not the focus of the current manuscript. That being said, we are working on coupling the P-SKRIPS model with the CICE and hope this will bring landfast ice in the new set up.

The Willmott index used in Table 2 is a rather unusual one, consider detailing its meaning in the section on methods. Also, this acronym is introduced before being cited.

We will add details of the Willmott index meaning in the methods section and move the acronym where it is cited.

Excerpt from the introduction of Willmott, Robeson, and Matsuura (2012):*Numerical models of climatic, hydrologic, and environmental systems have grown in number, variety and sophistication over the last few decades. There has been a concomitant and deepening interest in comparing and evaluating the models, particularly to determine which models are more accurate (e.g. Krause et al., 2005). Our interest lies in this arena; that is, in statistical approaches that can be used to compare model produced estimates with reliable values, usually observations.*

Subsection 2.3 is quite dense as a single paragraph; we recommend splitting it into several shorter paragraphs, for instance one for each type of dataset.

We agree, this section is indeed one single paragraph and wee will rewrite it and split it into shorter paragraphs.

**Other remarks**

The abstract is very short and does not render the breadth of the findings of this study. The authors should consider including more context, more results, and more perspectives. We will expand the abstract to present the context, perspectives and results more explicitly.

Writing: The writing is in general impeccable here is the few typos we found:

*We thank you for commenting on our writing. We will fix all the typos liste below*

L44: )) → )
L103: )) → ) to times
L104: )) → )
L103: start → stars (yellow) and stars → (purple)

*The figure will be updated as only one set of moorings will be retained in the new version of the manuscript (DITx).*

L106: )) → )
L113: )) → )
L154: "larger" than what ?

*Than for the SMOS satellite data. We will correct this sentence.*

L189: man → many?

*Yes, this will be corrected*

L191: need a space between "figure 5" and "as" otherwise it's confusing.
L200: $W.m^{-2}$ is missing, also a ";" is missing after "heat"; on that line, use the proper symbol for +/-
L202: remove "and" after "wind"
L222: )) → )
L339: displays → display
L359: )) → )

**Suggestions:**

L6 We propose "coastal polynyas activities" instead of "polynya activity".

*We will remove references to polynya extent and polynya activity, and will rephrase this sentence as "to simulate the atmosphere, sea ice and ocean in the Ross Sea, with a focus on Terra Nova Bay.*

L49: You could directly mention here which are the coastal polynyas instead of later as in L52.

*Yes, we will correct this.*

L158: We do not understand the sentence "Satellite products also show insufficient variation over the year.".

*Yes, we will rephrase the sentence.*

L166: "AR" is never defined.

*Apologies, we will fix this in the new version of the manuscript.*

L217: the acronym "WI" is used but only introduced in a table later; consider expanding it here too.

*THhank you, we will introduce the WI in the methods section.*

L411: Maybe use "realistic coastal polynyas activities" with "realistic polynyas" to be coherent with the rest of the paper.

*We will rephrase this sentence, as well as refine all references to polynyas, polynya extent and activity.*

**Figures**

Figure 1: Change "simulated domain" to "domain of simulation" since a domain is not simulated - In panel c, consider adding the text of the different items (red box, red star, green start, etc.) directly in the figures near

the symbols, this would be much more visual.

We will update the figure (and remove the location of the LDEO and MORSEA moorings) and the caption in the new version of the mansucript.

Figure 2: We would propose to change the color of the lines to make the distinction between satellite, reanalysis, and model output. Cryosat, AMSR, NOAA, SSMI and Bootstrap would go in shades of blue; GLORYS and ORA would go in shades of red; and the three sims would go in shades of green (for example). The use of dash vs solid lines is a bit confusing - Alternatively, consider using a shading to display the range of observations.

THank you for this suggestion, we will update the figure. We will also remove sim002 and sim003 from the graph.

Figure 5: There is a "a)" label, but no figure panel corresponds to it.

Thank you, we will fix this.

Figure 6: We suggest merging it with Figure 5, as both figures share the same data - If we get it right, in the last sentence: the AMPS dataset should be indicated in blue (not pink), and conversely the AWS data should be shown in pink (not blue).

Thank you, we will move Figure 5 to the supplementary material and only retain Figure 6 in the main text. And yes, the caption will be corrected.

Figure 7-8: Axis labels are not always necessary on every panel; we suggest keeping them only on the outer edges of the figures. This recommendation could also apply to other figures.

These figures will be removed from the new version of the manuscript, and will update the other figures to increase readability and clarity.

Figure 9-10-11: We suggest merging these figures, as they provide similar information, and moving some of the panels to the Supplementary Material.

As both reviewers commented on the length of the manuscript and the amount of observational datasets used in this study, we decided to only retain Figure 9 (DITx moorings) in the new version of the mansucript. We feel that this is appropriate, as the signal indicated by the other moorings is very similar to that visible in figure 9.

Figure 12-13: We are not clear why the bathymetry of the model and the references are so different. Can the authors clarify this? - The legend is not sufficiently detailed or explicit.

The bathymetry datasets vary in the different datasets, as well as the resolution at which the data is supplied. All of the sections were plotted for the southernmost location, standing in for the ice shelf front (as was displayed in figure S1 in the Supplementary material of the preprint). We agree that the Ross bank and the deeper area of the Ross Sea Polynya are better represented in the Bedmap2 dataset, used in the P-SKRIPS model. We will add more detail in the caption.

**References**

Fons, S., N. Kurtz, and M. Bagnardi (2023). "A decade-plus of Antarctic sea ice thickness and volume estimates from CryoSat-2 using a physical model and waveform-fitting". In: The Cryosphere 17, pp. 2487–2508. https://doi.org/https://doi.org/10.5194/tc-17-2487-2023.

Fretwell, P. et al. (2013). "Bedmap2: improved ice bed, surface and thickness datasets for Antarctica". In: The Cryosphere 7.1, pp. 375–393. https://doi.org/10.5194/tc-7-375-2013. https://tc.copernicus.org/articles/7/375/2013/.

Hersbach, H. et al. (n.d.). "The ERA5 global reanalysis". In: Q. J. Roy. Meteor. Soc. 146 (), pp. 1999–2049. `https://doi.org/https://doi.org/10.1002/qj.3803,2020`.

Kern, S., B. Ozsoy-Çiçek, and A.P. Worby (2016). "Antarctic Sea-Ice Thickness Retrieval from ICESat: Inter-Comparison of Different Approaches". In: Remote Sens. 538. `https://doi.org/https://doi.org/10.3390/rs8070538`.

Locarnini, R. A. et al. (2018). Temperature. Ed. by A. Mishonov Technical Ed. Vol. 1. NOAA Atlas NESDIS 81, 52pp.

Malyarenko, M. et al. (2023). "Conservation of heat and mass in P-SKRIPS version 1: the coupled atmosphere-ice-ocean model of the Ross Sea". In: Geoscientific Model Development 16, pp. 3355–3373. `https://doi.org/https://doi.org/10.5194/gmd-16-3355-2023`.

Padman, Laurie et al. (2002). "A new tide model for the Antarctic ice shelves and seas". In: Annals of Glaciology 34, pp. 247–254. `https://doi.org/10.3189/172756402781817752`.

Willmott, C.J., S.M. Robeson, and K. Matsuura (2012). "A refined index of model performance". In: Int. J. Climatol. 32, pp. 2088–2094. `https://doi.org/https://doi.org/10.1002/joc.2419`.

Zhang, H., D. Menemenlis, and I. Fenty (2018). ECCO LLC270 Ocean-Ice State Estimate. dataset. `https://dspace.mit.edu/handle/1721.1/119821?show=full`.